# Application of Liquid Chromatography/Tandem Mass Spectrometry for Quantitative Analysis of Plasmalogens in Preadolescent Children—The Hokkaido Study

**DOI:** 10.3390/diagnostics15060743

**Published:** 2025-03-16

**Authors:** Yifan Chen, Siddabasave Gowda B. Gowda, Divyavani Gowda, Jayashankar Jayaprakash, Lipsa Rani Nath, Atusko Ikeda, Yu Ait Bamai, Rahel Mesfin Ketema, Reiko Kishi, Hitoshi Chiba, Shu-Ping Hui

**Affiliations:** 1Faculty of Health Sciences, Hokkaido University, Kita-12, Nishi-5, Kita-ku, Sapporo 060-0812, Japan; feifeierfan@gmail.com (Y.C.); gowda@gfr.hokudai.ac.jp (S.G.B.G.); divyavani@hs.hokudai.ac.jp (D.G.); aaraki@cehs.hokudai.ac.jp (A.I.); krahel@cehs.hokudai.ac.jp (R.M.K.); 2Graduate School of Global Food Resources, Hokkaido University, Kita-9, Nishi-9, Kita-ku, Sapporo 060-0809, Japan; jayashankar.jayaprakash.q1@elms.hokudai.ac.jp (J.J.); lipsarani.nath.y6@elms.hokudai.ac.jp (L.R.N.); 3Center for Environmental and Health Sciences, Hokkaido University, Kita-12, Nishi-7, Kita-ku, Sapporo 060-0812, Japan; u-aitbamai@med.hokudai.ac.jp (Y.A.B.); rkishi@med.hokudai.ac.jp (R.K.); 4Department of Nutrition, Sapporo University of Health Sciences, Nakanuma Nishi-4-2-1-15, Higashi-ku, Sapporo 070-0894, Japan; chiba-h@sapporo-hokeniryou-u.ac.jp

**Keywords:** plasmalogens, liquid chromatography, mass spectrometry, children’s plasma, overweight

## Abstract

**Background**: Plasmalogens (Pls) are phospholipids with a unique structure, abundant in the brain and heart. Due to their chemical instability and analytical difficulties, less information is available compared to other phospholipids. The importance of Pls in several cellular processes is known, one of which is their protective effect against oxidative damage. The physiological role of Pls in human development has not been elucidated. Despite their clinical importance, the quantitative analysis of Pls in children’s plasma has been limited. **Methods**: This study aims to determine the plasma levels of Pls in prepubertal children using liquid chromatography/tandem mass spectrometry (LC-MS/MS). The plasma samples used were obtained from 9- to 12-year-old girls (n = 156) and boys (n = 178), n = 334 in total, who participated in the Hokkaido study. **Results**: Ethanolamine plasmalogen (PlsEtn) and choline plasmalogen (PlsCho), both carrying eicosapentaenoic acid, were significantly lower in girls than in boys. In both sexes, the plasmalogen levels for the 12-year-old children were lower than those for the 9-year-old children. PlsCho (16:0/18:2) was lower in the overweight children than in the normal-weight children for both sexes. PlsEtn (18:0/20:4) was the most abundant ethanolamine-type plasmalogen in both sexes. **Conclusions**: This study is the first report on plasmalogen levels and molecular types in children’s plasma. This study provides the information needed to understand the role of Pls in human developmental processes and may open up new opportunities in the future to control age-related changes in Pls.

## 1. Introduction

Glycerophospholipids (GPs) are major membrane lipids that are further classified into different classes depending on the diversity of their chemical structure. GPs in which the viny-ether alkyl chain is attached to the sn-1 position (R1 in Figure 1a) of a glycerol backbone and the acyl chain is attached at the sn-2 position are coined as plasmalogens (Pls). In general, Pls are restricted to a variety of fatty acids (FAs), including polyunsaturated fatty acids (PUFAs) at the sn-2 position (R2 in Figure 1a) and the saturated or monounsaturated alkenyl chain at the sn-1 position. The FAs of Pls at the sn-1 position are typically restricted to C16:0 (palmitoyl), C18:0 (stearoyl), or C18:1 (oleoyl) carbon chains [1,2]. In humans, Pls can make up to 18% of the total mass of GPs, but these are distributed unevenly across different types of tissues and cells [3]. Pls are further grouped depending on the polar head moiety attached to the sn-3 position. If the phosphoethanolamine is attached to the sn-3 position, they are named ethanolamine plasmalogens (PlsEtns), whereas if the phosphocholine is attached to the sn-3 position, then they are named choline plasmalogens (PlsChos), as shown in Figure 1a. Both PlsEtns and PlsChos are abundant in tissues such as skeletal muscle, the brain, lungs, and kidneys [1,4]. In the human brain, 58% of phosphatidylethanolamines (PEs) are PlsEtn, whereas in the human heart, 26% of PEs are PlsCho [5,6]. Pls play a crucial role in the systemic development and maintenance function of the central nervous system [7]. Over the past two decades, the studies focusing on Pls and their analysis have increased due to their various roles in health and disease management. The decreased levels of Pls have been implicated in a range of neurological disorders, such as Alzheimer’s disease (AD), gastrointestinal cancer, and cardiovascular and respiratory diseases [6,8,9,10,11]. Moreover, studies have also shown that plasmalogen deficiency is more common in children with certain medical conditions, such as cystic fibrosis and autism spectrum disorder [12]. Therefore, there is great scope for research on the quantitative determination of Pls in children and adolescent health.

Pls in serum samples from female adolescents with and without pain were investigated by global lipidomics [13]. In another study, gas chromatography was applied to determine Pls in the red blood cells of neonates and 1–8-year-old children [14]. Liquid chromatography/tandem mass spectrometry (LC-MS/MS) is widely applied for the determination of Pls in human plasma samples due to its high sensitivity and selectivity [15,16]. A previous study showed that the concentration of Pls in human plasma was 0.5–13.6 µM [17]. Despite their clinical significance, their absolute levels in children’s plasma are not yet fully determined or have only been used in a limited sample size. As for our knowledge, no studies have been conducted to determine the concentration of Pls in preadolescent children of different ages, genders, and percentages of body weight (POW). In this study, we aim to analyze the absolute concentration of Pls in plasma samples collected from 334 preadolescent children (from 9 to 12 years old) of the Hokkaido study using targeted LC-MS/MS, as shown in Figure 1a [18,19,20,21]. Furthermore, in this paper, the variations in Pls by age, gender, and POW are discussed.

## 2. Materials and Methods

The overall study design is depicted in Figure 1b. High-purity solvents such as methanol (LC/MS grade, purity: 99.8%) and chloroform (high-performance liquid chromatography grade, purity: 99.7%) were purchased from Kanto Chemical Co., Inc. (Tokyo, Japan). Butylated hydroxytoluene (BHT) and ammonium acetate were secured from Sigma-Aldrich (St. Louis, MO, USA). The internal standards of Pls (PlsEtn-p16:0/17:0 with the concentration 106 nmol/mL and PlsCho-p16:0/17:0 with the concentration 110 nmol/mL) were synthesized by the earlier reported method from our laboratory [15].

The non-fasting plasma samples of children (n = 334) from 9 to 12 years old were collected during the Hokkaido study and stored at −80 °C until further analysis.

The Hokkaido study of Environment and Children’s Health is a study that began in 2002 involving two birth cohorts: (a) the Toho hospital cohort with one obstetric hospital in Sapporo City, and (b) the Hokkaido large-scale cohort with 37 hospitals and clinics in the Hokkaido prefecture. The details about the Hokkaido study were provided in our previously published literature [7,18,19,20,21,22,23]. The prior approval was taken from the Hokkaido University ethical committee of the Department of Health Science, and the approval number is 22–33. In the extraction protocol, 100 µL of a plasma sample was taken into an Eppendorf tube and extracted by the modified Folch method [24,25]. Specifically, 400 µL of methanol was added to 100 µL of a plasma sample in a 1.5 mL Eppendorf^®^ Tube, where 100 µL of methanol containing 0.002% BHT, and 100 µL of methanol containing 1 nmol/mL was used as the internal standard mixture (PlsCho p16:0/17:0 and PlsEtn p16:0/17:0). The extraction of Pls was conducted twice with chloroform. The detailed extraction workflow is shown in Figure 1b. The extraction recoveries for both PlsEtn and PlsCho were found to be >80%.

A prominence HPLC (Shimadzu Corp., Kyoto, Japan) system connected with a TSQ Quantum Access Max mass spectrometer (Thermo-Fisher Scientific Inc., Waltham, MA, USA) was used to perform the LC/MS analysis of Pls. The Hypersil GOLD column (50 × 2.1 mm, 1.9 μm, Thermo Fisher Scientific Inc., MA, USA) was used for the chromatographic separation of Pls. The following conditions were used for separation: Pump A: methanol/water 5:1 *v/v* (with 10 mM ammonium acetate); Pump B: methanol, flow rate: 200 µL/min, the gradient elution: 0–2 min, 65% A, 45.0% B; 2–8 min, 20% A, 80% B; 8–10 min, 20% A, 80.0% B; 10–12 min, 65% A, 45% B and re-equilibration for an additional 3 min. About 10 µL of the standard/sample was set for injection. The mass spectrometer parameters for the analysis were set as follows: electron spray ionization (ESI) in the negative mode with a spray voltage of 4 kV, capillary temperature: 300 °C, and N2 sheet gas and auxiliary gas pressure; 40 psi and 55 psi. The raw data were processed by Xcalibur 2.2 (Thermo-Fisher Scientific Inc.). The quantification was performed by establishing the linearity curves for each Pl standard. The plasmalogen plasma concentration results are summarized in the Appendix A.

In this research, the statistical analysis was conducted by R studio ( R software with the version 4.1.0), and all the figures were made using GraphPad Prism 8.0.1 (GraphPad Software Inc., La Jolla, CA, USA). Student’s *t*-test and the Wilcoxon test were used to compare gender groups with the normality of the sample and equality of sample variance. In addition, two-way ANOVA analysis with aligned rank transformation (ART) and the Kruskal–Wallis rank sum test was conducted on other group comparisons based on the normality of the sample and equality of sample variances. Statistical significance was set at *p* < 0.05. The results were expressed as the mean ± standard deviation.

## 3. Results

### 3.1. Sex-Specific Differences in Plasmalogen Levels in Children’s Plasma

The quantitative levels of PlsEtn and PlsCho detected in children’s plasma are shown in Figure 2a. The results clearly show that arachidonic acid (ω-6)-containing PlsEtn 18:0/20:4 is the most abundant Pl species in the plasma of both boys and girls. Pls have multiple roles, including functions as endogenous antioxidants, maintaining cellular membrane integrity, and they are also involved in cholesterol trafficking [1]. In our results, most of the Pls were found to have no significant differences between sexes and are quite consistent with a previous report, which showed a similar plasma lipid profile between men (27–33 years) and women (26–33 years) [26]. However, eicosapentaenoic acid (EPA, ω-3) containing PlsEtn 16:0/20:5 and PlsCho 16:0/20:5 was significantly lower in girls compared to boys. To verify the main age-group contributor for decreased eicosapentaenoic acid Pls in girls compared to boys, we performed the sex-and age-dependent analysis for eicosapentaenoic acid Pls. The results are shown in Figure 2b. There was no significant difference observed among the 9, 10, 11, and 12-year-old groups between girls and boys in their levels of PlsEtn 16:0/20:5. However, a significant decrease in the levels of PlsCho 16:0/20:5 was observed for 9, 10 and 12-year-old groups of girls compared to boys.

Moreover, to verify the gender and age effect on Pls-Etn16:0/20:5, a two-way ANOVA analysis with ART was conducted on the sex-specific groups. Compared with the *t*-test and Wilcoxon test, two-way ANOVA with ART showed a more direct interpretation of the gender effect on plasmalogen groups. From the results, PlsEtn 16:0/20:5 is significantly affected by the gender factor (*p*-value = 0.015), while PlsCho 16:0/20:5 is significantly affected by the age factor (*p*-value = 0.003) and gender factor (*p*-value = 0.005). Furthermore, the other Pl types are only significantly influenced by the age factor with *p*-values less than 0.05. Thus, the PlsEtn 16:0/20:5 concentration caused by gender difference is considered a potential interpretation of the difference in metabolism between girls and boys.

### 3.2. Age-Dependent Changes in Plasmalogen Levels in Children’s Plasma

The analysis of Pls in the plasma of 9- to 12-year-old children of both sexes was performed. The age-dependent variation in Pls in boys and girls is shown in Figure 3a and Figure 3b, respectively. Among the ethanolamine plasmalogens, PlsEtn 16:0/18:2, 16:0/18:1, 18:0/22:6, 18:0/20:4, and 18:0/18:2 were shown to be significantly lower in the plasma of 12-year-old compared to 9-year-old boys. Among choline plasmalogens, PlsCho 16:0/20:5 and 16:0/18:2 were also significantly lower in 12-year-old compared to 9-year-old boys. However, in girls, almost all the detected Pls including PlsEtn (16:0/20:5, 16:0/18:2, 16:0/18:1, 18:0/22:6, 18:0/20:4, 18:0/18:2) and PlsCho (16:0/20:5, 16:0/18:2, 16:0/18:1) were significantly lower in 12-year-old compared to 9-year-old girls. Although the age-dependent variation in Pls is not uniform, a significantly lower amount in 12-year-old children suggests Pl loss with aging. In our previous study, PlsCho 16:0/18:2 was determined in adult plasma (20 years old), and the value was 4 pmol/μL, which further supports our assumption that Pls may decrease with aging [15].

### 3.3. Change in Plasma Plasmalogens in Overweight Children’s Plasma

The percentage of overweight (POW) individuals is widely used to define childhood obesity in Japan and is established on age- and sex-specific standard body weight for height. POW was calculated as [measured weight (kg) standard weight (kg)/standard weight (kg)] × 100. Children with POW < −20% and children with POW ≥ +20% were classified as underweight and overweight, respectively. Children with POW between these cutoff points were categorized as having a normal weight [27]. The changes in plasma Pls levels between underweight, normal, and overweight boys and girls are shown in Figure 4a and Figure 4b, respectively. Among boys, there was no significant difference in the Pl levels between normal and overweight children except for PlsCho 16:0/18:2 and PlsCho 16:0/18:1. They appear to be significantly lower in the plasma of overweight boys compared to normal-weight boys. Interestingly, similar results were observed in girls; PlsCho 16:0/18:2 and PlsCho 16:0/18:1 also decreased in overweight girls compared to normal-weight girls. In addition, PlsCho 16:0/18:1 also decreased in overweight girls compared to the underweight subgroup. Decreased linoleic acid-derived and oleic acid-derived choline Pls appear to be lipid biomarkers for obesity in both sexes.

## 4. Discussion

In this study, we analyzed PlsEtn and PlsCho types. Their distribution in the human body varies significantly in different tissues. In mammalian cells, the head groups of Pls are almost exclusively ethanolamine (PE-Pls) and choline (PC-Pls). Pls are distributed in various tissues and compartments throughout the body, including the brain, heart, kidneys, muscles, and blood. In plasma, PE-Pls and PC-Pls account for 50% of total phosphatidylethanolamine (PE) and 5% of total phosphatidylcholine (PC), respectively [3]. Moderate amounts of Pls are found in skeletal muscle and blood cells. In addition, the liver has the lowest content of Pls [11].

Our analysis results showed PlsEtn with EPA in the sn-2 position were found to be significantly different between genders (PlsEtn 16:0/20:5). EPA is a type of ω-3 FA as well as an essential fatty acid that plays a crucial role in brain function and mental health. There is increasing evidence suggesting that EPA may have a beneficial effect on depression. One of the most prominent theories is that inflammation plays a role in depression. ω-3 FAs, particularly EPA, have anti-inflammatory properties, and some studies have suggested that individuals with depression may be influenced by inflammatory markers. EPA has been shown to reduce pro-inflammatory cytokines in the body, which might help alleviate symptoms of depression by reducing the inflammatory processes that are thought to contribute to mood disorders [28]. Moreover, recent research has indicated that EPA may help promote neuroplasticity, which can support recovery from mental health conditions like depression [29]. Recently, a similar study in adolescents found no association between red blood cell DHA levels and depression severity or oral memory performance but did find a difference with red blood cell EPA [30,31]. Previous sexual differentiation in cognitive and behavior studies suggested that depression is more frequent in females than in males, while Dopamine system disorders, such as Parkinson’s disease and schizophrenia, are more common in the male group than females [32,33]. In a previous study, females had higher DHA-containing Pls in plasma compared to males [34]. Thus, EPA- and DHA-containing Pls might be a factor influencing different cognitive disorders. In our study, EPA-containing Pls, which are correlated with depression, were found less in girls’ plasma than in boys’ plasma; DHA-containing Pls, which are correlated with Parkinson’s disease, was not significantly different between genders. Therefore, our study might be a potential explanation for depression concerning the phenomena mentioned above. However, the results of ω-3 FAs and brain function in adolescents, especially depression, remain uncertain because one intervention trial in a large group of healthy adolescents with krill oil indicated no significant improvement in depressive symptoms, while an intervention trial in 60 depressed adolescents found significant improvement [35,36]. Many neuroscientists insist that environmental factors and socioeconomic status should be considered in sex-differential cognitive behavior as well [32].

For the results of PlsCho concentrations between genders, only EPA-containing PlsCho showed a significant difference (*p* < 0.001). PlsCho types play an important role in cardiac tissue but represent a minor species in most other organs [9]. Previous research demonstrated that several oxidative fatty acid metabolites showed differences in males’ and female’s serum or plasma [26]. Another study in Canada proved that the sex difference in mRNA also influenced the beta-oxidation pathway by affecting the major enzyme activities and indicated that women have a higher capacity for medium-chain fatty acids (MCFAs) and utilization of long-chain fatty acids (LCFAs) in skeletal muscle during endurance exercise. The study demonstrated that one of the possible contributing factors for the observed sex differences could be estrogen effects. Previous studies have suggested that estrogen might play an important role in regulating substrate utilization [37]. The result might demonstrate that women have greater MCFAs and LCFA transport capabilities in skeletal muscle and provide one possible contributing factor, estrogen, for the observed Pl concentration difference between boys’ and girls’ plasma.

In comparison, among ages, the variations in Pls (excluding PlsEtn 16:0/20:5) are influenced by age factors. AA-containing PlsEtn and DHA-containing PlsEtn significantly decreased from 9-year-old children to 12-year-old children. These findings may indicate that there is a potential decreasing trend with aging from youth for PlsEtns, which play crucial roles in protecting the brain from neurodegenerative diseases. A previous study involving 148 elderly participants (>65 years old) showed that serum Pls are positively correlated with high-density lipoprotein and decrease with aging [38]. The association between a Pl deficiency and various neurodegenerative diseases, peroxisome diseases, and heart diseases was extensively reviewed [39]. Furthermore, we found that Pls decline in girls’ plasma in a more drastic trend than those in boys’ plasma from Figure 4a,b. As for this phenomenon, we suppose that the contributing factor might be estrogen as well. The difference between genders becomes more pronounced with increasing age as girls go through puberty.

In the comparison among overweight children, the results indicate that there is a significant decrease in PlsChos 16:0/18:1 and 16:0/18:2 in overweight children. The same relationship between PlsChos and body shape was published by a previous study, which indicated that PlsChos 16:0/18:1 and 16:0/18:2 have a negative relationship with waist circumference. One recent study demonstrated that a high-fat diet induces TMEM86A, a putative lysoplasmalogenase, which overexpress and leads to decreasing Pls [40]. In a previous study, plasmalogen loss was linked to the remodeling deficiency in mitochondria [41]. Triacylglycerols were observed to be increased in the plasma of obese children and adolescents [42], whereas negative correlations were observed between plasma Pls and serum triacylglycerols [43].

In our previous study, we showed that reduced-serum lipoprotein PlsEtn in patients with non-alcoholic steatohepatitis due to oxidation [44] and serum PlsCho was significantly lower in males with significant coronary stenosis [45,46]. The concentration of β-amyloid was noted to be negatively related to PlsEtns in Alzheimer’s disease (AD) patient serum samples [47]. Thus, being overweight, which is partially caused by cholesterol accumulation, was implied to be related to PlsEtns variation among AD patients. Such notation, on the contrary, might indicate that being underweight does not influence the concentration of Pls significantly. In addition, in this study, the results imply that being overweight might be negatively associated with some Pl choline types. However, we are not sure about the POW results of the underweight boy participants due to the low number (n = 3) included for accurate statistical analysis. In addition, our study involved non-fasting preadolescent children; thus, the variations in the plasma lipidome of overweight children could be influenced by their diet. However, recent research guidelines suggest using measurements of fasting lipid profiles of obese children rather than non-fasting samples, which limits our study.

However, we should note that Pls are not only affected by endogenous factors but also exogenous factors. In previous research, dietary Pl was considered to be a factor influencing the concentrations of Pls [2,48]. In this study, the maximum, minimum, and mean values of the concentrations of individual Pl species are shown in Table 1. From this table, we can see that PUFA-containing PL (PlsEtn 16:0/20:5, PlsEtn 18:0/22:6, PlsEtn 18:0/20:4 and PlsCho 16:0/18:2) concentrations are higher than other Pl concentrations. PUFAs are essential FAs that cannot be synthesized by human beings; they must be obtained by dietary intake. Specifically speaking, PUFAs 20:4, 20:5, and 22:6 are generally obtained from fish, meat, and eggs, while PUFA 18:2 is mainly obtained from plant oil [49]. Thus, the loading values of PUFA-containing Pls in Table 1 suggest that dietary plasmalogen might be a possible contributing factor to the concentration of Pls. In summary, we suppose that the differences in LCFAs could reflect metabolism (with the synthesis of Pls included) between men and women (with the feminization of dietary habits included).

This study has many limitations as follows: (1) non-fasting plasma samples were collected from the children’s plasma and used for analysis; since diet can influence Pl levels, the result may be affected by the sample collection method. (2) Physical activity was also not monitored, which could have an influence on plasma levels, and (3) a limited number of plasmalogen molecular species were quantified in this study. A follow-up study may be essential to determine the plasma Pls in the same population when they are adolescents.

## 5. Conclusions

Overall, a targeted LC-MS/MS method for the sensitive determination of Pl levels in children’s plasma samples was applied. In this Hokkaido study, more than 330 children were involved, and their plasma samples were collected. In this study, the EPA-containing PlsEn and PlsCho were lower in girls compared to boys, which might be caused by estrogen. Moreover, the Pls (except for PlsEtn 16:0/20:5) were depleted with aging, as suggested by lower Pls in 12-year-old children compared to 9-year-old children. With increasing age, the decrease in Pls is more severe. In addition, the Pl concentration difference in the group of girls appeared to be more drastic than those in boys due to the total numbers of significantly different pairs in each gender. Such a phenomenon may indicate the estrogen effect on Pls and an initial declining trend in Pls. Furthermore, decreased Pls (PlsCho16:0/18:2 and Pls 16:0/18:1) in the plasma of overweight children were observed in both sexes. Finally, the loading values of the Pl concentration suggested that the amount of Pls in plasma is influenced by dietary intake since the oral intake of Pls might protect us against some disorders. As for our knowledge, this is the first Hokkaido study on the determination of absolute levels of Pls in children’s plasma, considering age, gender, and POW. This study will help us to understand the importance of Pls in children’s health management comprehensively.

## Figures and Tables

**Figure 1 diagnostics-15-00743-f001:**
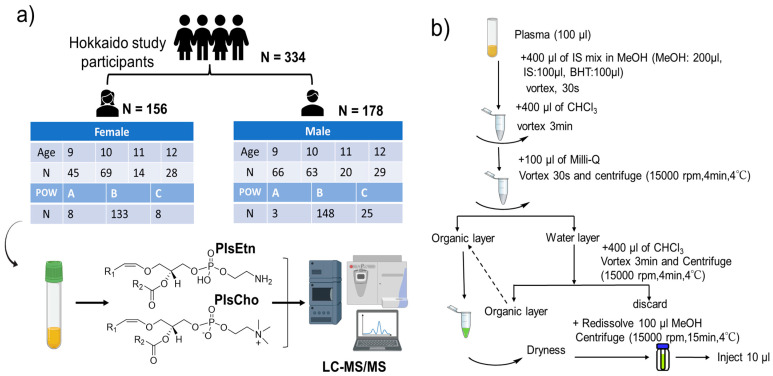
(**a**) The sample information used for plasmalogen analysis. The number of participants is 334 (N_Boy_ = 178, N_Girl_ = 156); the percentage of weight (POW) contains three subgroups (A: underweight group, B: normal weight group, and C: overweight group). The age group contains four subgroups (the 9-year-old group, 10-year-old group, 11-year-old group, and 12-year-old group). (**b**) Workflow of the extraction protocol for plasmalogens.

**Figure 2 diagnostics-15-00743-f002:**
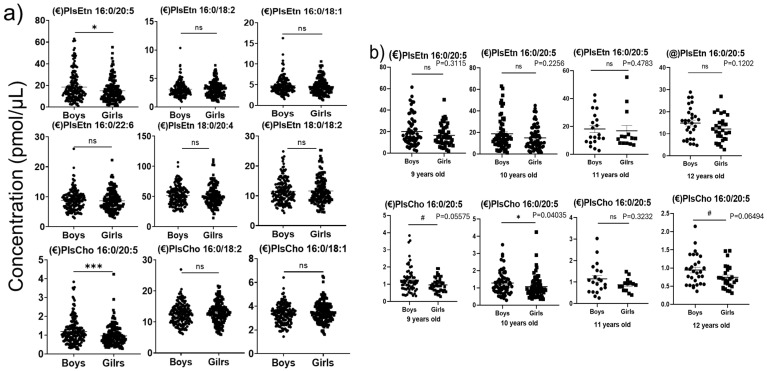
(**a**). Sex-dependent changes in plasmalogen concentrations in children’s plasma. (**b**). Sex- and age-dependent variations in PlsEtn 16:0/20:5 and PlsCho 16:0/20:5 concentrations in children’s plasma. (Wilcoxon test (€) and *t*-test (@); *p*-value: (ns: non-significant, *p* < 0.1 (#), *p* < 0.01 (*), *p* < 0.001 (***)).

**Figure 3 diagnostics-15-00743-f003:**
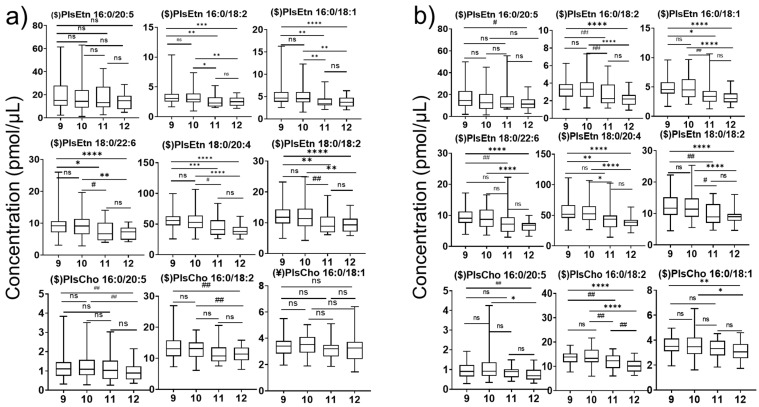
(**a**) Boxplot with min to max values for age-dependent changes in plasmalogen concentrations in boys’ plasma. (**b**) Boxplot with min to max values for age-dependent changes in plasmalogen concentrations in girls’ plasma (Kruskal–Wallis ($) and one-way ANOVA (¥) analysis; *p*-value: (ns: non-significant, *p* < 0.1 (#), *p* < 0.05 (##), *p* < 0.01 (*), *p* < 0.005 (**), *p* < 0.001 (***), *p* < 0.0005 (****)).

**Figure 4 diagnostics-15-00743-f004:**
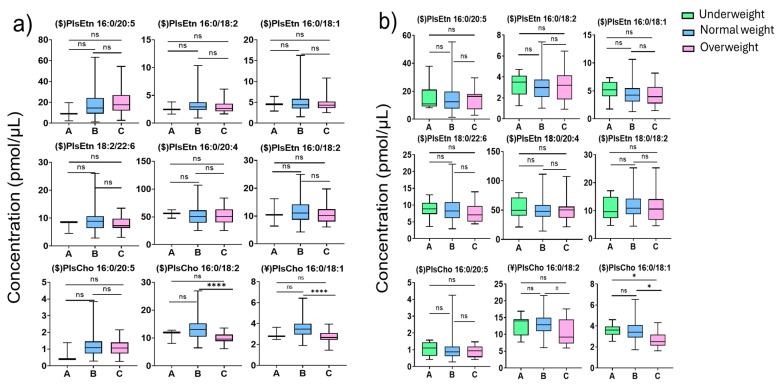
(**a**) Boxplot with min to max values for POW-dependent changes in plasmalogen concentrations in boys’ plasma. (**b**) Boxplot with min to max values for POW-dependent changes in plasmalogen concentrations in girls’ plasma (A: underweight; B: normal weight; C: overweight; Kruskal–Wallis ($) and one-way ANOVA (¥) analysis; *p*-value: (ns: non-significant, *p* < 0.1 (#), *p* < 0.01 (*), *p* < 0.0005 (****)).

**Table 1 diagnostics-15-00743-t001:** Amount of plasmalogens in girls’ and boys’ plasma (pmol/μL). Max means the maximum values; min means the minimum values; M means the mean values; and SD means the standard deviation.

	PlsEtn (pmol/uL)						PlsCho (pmol/uL)		
**Boys**	16:0/20:5	16:0/18:2	16:0/18:1	18:0/22:6	18:0/20:4	18:0/18:2	16:0/20:5	16:0/18:2	16:0/18:1
MAX	63.06	10.37	16.27	26	106.68	24.89	3.84	26.91	6.42
MIN	1.17	0.94	1.53	2.87	25.4	4.26	0.26	6.12	1.44
M ± SD	18.57 ± 13.14	3.11 ± 1.28	4.77 ± 1.95	8.78 ± 3.25	51.85 ± 15.36	10.84 ± 3.99	1.20 ± 0.67	12.58 ± 3.33	3.34 ± 0.79
**Girls**	16:0/20:5	16:0/18:2	16:0/18:1	18:0/22:6	18:0/20:4	18:0/18:2	16:0/20:5	16:0/18:2	16:0/18:1
MAX	55.31	7.33	10.63	22.22	111.13	25.33	4.24	21.59	6.52
MIN	1.43	0.92	1.29	2.94	14.24	4.48	0.29	5.93	1.62
M ± SD	15.14 ± 10.16	3.12 ± 1.29	4.56 ± 1.89	8.65 ± 3.24	0.73 ± 17.04	11.64 ± 4.31	0.97 ± 0.52	12.941 ± 3.41	3.48 ± 0.86

## Data Availability

The relevant raw data files for this study are accessible publicly from the following link: https://data.mendeley.com/drafts/dbznbj62dd (accessed on 25 October 2024).

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
