# Peer review of "Application of Liquid Chromatography/Tandem Mass Spectrometry for Quantitative Analysis of Plasmalogens in Preadolescent Children—The Hokkaido Study"

_diagnostics, 2025, doi:10.3390/diagnostics15060743_

Round 1
Reviewer 1 Report
Comments and Suggestions for Authors
The title of the manuscript "Quantitative determination of plasmalogens by LC-MS/MS in preadolescent children-the Hokkaido study" did not actually reflects the results presented in the manuscript. It is unclear that the aim of the authors either to present Hokkaido study results or to present LC-MS/MS methodology along with its application results?. Authors discussed more of Hokkaido study results, than the LC-MS/MS methodology, development and validation. It is required to revise the title of the manuscript, provide the complete LC-MS/MS methodology (along with challenges during the method development, mass spec conditions m/z values, standards and QCs, Chromatograms etc.), and method validation results either in the manuscript or as supplemental data.
Author Response
Reviewer 1
Summary response : Thank you very much for taking the time to review this manuscript. Please find the detailed responses below and the corresponding revisions corrections highlighted in the re-submitted files
Comments 1: The title of the manuscript "Quantitative determination of plasmalogens by LC-MS/MS in preadolescent children-the Hokkaido study" did not actually reflect the results presented in the manuscript. It is unclear that the aim of the authors either to present Hokkaido study results or to present LC-MS/MS methodology along with its application results. Authors discussed more of Hokkaido study results, than the LC-MS/MS methodology, development and validation.
Response 1: Thank you very much for your valuable comments. Our study aim is to perform the application of the method that is already established in the laboratory (Reference 15: Anal Bioanal Chem 2011, 400(7), 1923–1931). Hence, we discussed more on the Hokkaido study results. Since method validation is already being published in the literature for the human plasma samples from our laboratory we haven’t focused on this part in the present study.
Comments 2: It is required to revise the title of the manuscript, provide the complete LC-MS/MS methodology (along with challenges during the method development, mass spec conditions m/z values, standards and QCs, Chromatograms etc.), and method validation results either in the manuscript or as supplemental data.
Response 2: Thank you for your suggestion to revise the title. Considering the reviewer suggestion we have revised our title as “Application of LC-MS/MS for quantitative analysis of plasmalogens in preadolescent children-the Hokkaido study”. Additionally, we have provided the QC and sample/standards EICs in the supporting information cited in supplementary material part. Since, detailed mass spec conditions are already described in our previous studies which are cited here we omitted such details.
Reviewer 2 Report
Comments and Suggestions for Authors
The manuscript titled “Quantitative Determination of Plasmalogens by LC-MS/MS in Preadolescent Children: The Hokkaido Study” is well-researched and well-presented. However, the following minor comments should be addressed to further enhance the quality of the proposed research work.
- The abstract and introduction sections were well presented.
- In Section 2, please include the purity, make, and grade details of the chemicals and standards used in the proposed study.
- Consider subdividing this section into the following subsections: Chemicals and Reagents, Materials and Methods, and other relevant sections, as applicable.
- The presented ANOVA calculations and statistical approaches were quite impressive.
- The paragraphs in the discussion section are too lengthy; it is recommended to use subsections to explain them.
- The conclusion is well presented.
- Overall, the proposed work is quite impressive and well-studied.
Author Response
Reviewer 2
The manuscript titled “Quantitative Determination of Plasmalogens by LC-MS/MS in Preadolescent Children: The Hokkaido Study” is well-researched and well-presented. However, the following minor comments should be addressed to further enhance the quality of the proposed research work.
Response summary : Thank you very much for your valuable comments. I revised the manuscript by your advice. Please find the detailed responses below and the corresponding revisions corrections highlighted in the re-submitted files
Comment 1 : The abstract and introduction sections were well presented.
Response 1 : Thank you very much for your valuable comments.
Comment 2 : In Section 2, please include the purity, make, and grade details of the chemicals and standards used in the proposed study.
Response 2 : We added the information methanol purity 99.8 % (LC/MS grade), chloroform purity 99.7% (HPLC grade). PlsEtn-p16:0/17:0 with the concentration 106nmol/mL and PlsCho-p16:0/17:0 with the concentration 110nmol/mL.
Comment 3: Consider subdividing this section into the following subsections: Chemicals and Reagents, Materials and Methods, and other relevant sections, as applicable.
Response 3 : Thank you very much for your advice. However, we cannot separate the material and method section into several subsections due to the Journal template.
Comment 4 : The presented ANOVA calculations and statistical approaches were quite impressive.
Response 4 : Thank you very much for your comments.
Comment 5 : The paragraphs in the discussion section are too lengthy; it is recommended to use subsections to explain them.
Response 5 : Thank you very much for your advice. We separated the paragraph by " Our analysis results showed PlsEtn with EPA in the sn-2 position were found significantly different between genders (PlsEtn 16:0/20:5). ".
Comment 6 : The conclusion is well presented.
Response 6 : Thank you very much for your kind comments.
Comment 7 : Overall, the proposed work is quite impressive and well-studied.
Response 7 : Thank you very much for your kind comments.
Round 2
Reviewer 1 Report
Comments and Suggestions for Authors
Thank you for responding my comments. I recommended to accept the manuscript.